# Analysis of the B-Type Natriuretic Peptide and the Aminoterminal-Pro-B-Type Natriuretic Peptide in Different Parrot, Raptor and Owl Species

**DOI:** 10.3390/vetsci9020064

**Published:** 2022-02-01

**Authors:** Anja Hennig, Lydia Mohr, Michael Fehr, Marko Legler

**Affiliations:** 1Department of Small Mammal, Reptile and Avian Diseases, University of Veterinary Medicine Hannover, Foundation, Bünteweg 9, 30559 Hannover, Germany; Michael.Fehr@tiho-hannover.de (M.F.); marko.legler@tiho-hannover.de (M.L.); 2Clinic for Poultry, University of Veterinary Medicine Hannover, Foundation, Bünteweg 9, 30559 Hannover, Germany; Lydia.Mohr@laves.niedersachsen.de

**Keywords:** brain natriuretic peptide, heart, cardiovascular diseases, parrots, raptors, owls

## Abstract

The B-type natriuretic peptide (BNP), a member of the natriuretic peptide family and a cardiac hormone, is produced mainly in the ventricular myocytes and released into the circulation due to mechanical stimuli during an increasing cardiac wall stretch. BNP has a significant role in the regulation of the cardiovascular system and body fluid. The concentration of this hormone and of the biologically inactive amino-terminal-prohormone in the blood plasma is a helpful diagnostic tool for detecting cardiovascular diseases in human medicine and can be used as a prognostic marker for the risk of mortality, whilst such a tool does not exist for avian medicine. To date, the amino acid sequence of BNP is not known for many of the species commonly presented in avian consultation. In this study, the amino acid sequence of BNP and the prepropeptide was described for 12 parrot species as well as 3 raptor and 3 owl species by polymerase chain reaction (PCR) after RNA isolation from the heart. The results showed a high similarity between the amino acid sequences in the mature peptide region of the BNP. The prepropeptide showed several differences between the examined species, some of them shared by closely related species.

## 1. Introduction

The Brain natriuretic peptide (B-type natriuretic peptide, BNP) belongs to the family of natriuretic peptides and was first isolated from the brain of a pig [1]. The natriuretic peptide family includes several different natriuretic peptides, which are all characterized by a 17 amino acid ring structure, formed by an intramolecular disulphide bond. However, the concentration of BNP is much higher in the heart. This peptide is synthetized in cardio myocytes and released into the blood stream as a cardiac hormone in response to an increased ventricular wall stress and to myocardial ischemia [2,3]. 

BNP seems to be the primary natriuretic peptide of the heart in birds, although a C-type-natriuretic peptide has also been isolated in cardiac myocytes. An A-type natriuretic peptide (atrial natriuretic peptide; ANP) could not be found in birds [4,5]. 

Several studies about the physiological effects of BNP in birds exist, most of them concerned with poultry, such as ducks and chicken [6,7,8,9]. The expression of the prepropeptide rises with an increasing cardiac wall stretch due to an increased blood volume and blood pressure [10]. The prepropeptide is known to consist of 140 amino acids in chickens. Within the cardiac myocytes, a pro-BNP, consisting of 116 amino acids, is split off from the prepropeptide from the signal sequence by a signal peptidase. During the release into the blood stream, the pro-BNP is enzymatically cleaved to the biologically active form, the BNP (mature peptide region), 29 amino acids long, and the amino-terminal-pro-BNP (NT-pro-BNP), so that both peptides can be found within the bloodstream [11]. Once released into the circulation, BNP works as an antagonist of the renin-angiotensin-aldosterone system [8]. It affects the kidneys, leading to an increased glomerular filtration rate and causes natriuresis and diuresis. The half-life time of BNP in the blood seems to be very short, being 1.2 min in Peking ducks [6,8,9]. 

In human medicine, BNP and NT-pro-BNP are used as diagnostic tools for detecting cardiac diseases, distinguishing them from other conditions causing similar symptoms. They can also be used as prognostic markers for the risk of cardiac events and their mortality in the future [12,13,14]. NT-pro-BNP has the benefit of a longer half-life time and long-term stability compared to BNP [15,16]. 

In pet birds, such a diagnostic tool would be very helpful too, since to date the diagnosis of diseases of the cardiovascular system has mostly been based on imaging methods [17,18]. To ascertain whether BNP or NT-pro-BNP might be potential candidates for use in diagnosing and monitoring cardiovascular diseases, further physiological studies would be required, especially in pet bird species. To our knowledge, no such studies exist, since the sequence of BNP has not yet been identified in many species commonly kept in private hands. The sequence of BNP is known, for example, for chickens and pigeons [4,19]. Previous studies showed that it is possible to detect the mature peptide region of BNP in grey parrots, which are common patients to the avian veterinarian [20]. However, the NT-pro-BNP was not examined in that study [21]. Many other species remain that have not been examined yet. 

Therefore, the aim of the present study was to examine the presence and sequence of BNP in different other species that might appear in avian medicine practice.

## 2. Materials and Methods

### 2.1. Birds, Samples and Ethical Statement

In this study, the hearts of 12 parrots, 3 raptors and 3 owls were investigated. The examined parrot species, all of them belonging to the order of *Psittaciformes*, included a scarlet macaw (*Ara macao*) and a blue-and-gold macaw (*Ara ararauna*), a solomons cockatoo (*Cacatua ducorpsii*) and a citron-crested cockatoo (*C. sulphurea citrinocristata*), a cockatiel (*Nymphicus hollandicus*), a peach-faced lovebird (*Agapornis roseicollis*) as well as a fischer’s lovebird (*Agapornis fischeri*), a budgerigar (*Melopsittacus undulatus*), an eastern rosella (*Platycercus eximius*), a red-lored amazon (*Amazona autumnalis lilacina*), and also a congo grey parrot (*Psittacus erithacus*) and timneh grey parrot (*Psittacus timneh*). The owls that were included in the study were a barn owl (*Tyto alba*), an eagle owl (*Bubo bubo*) and a tawny owl (*Strix aluco*), all representing the order of the *Strigiformes*. The raptors included two species representing the *Accipitriformes*, a hooded vulture (*Necrosyrtes monachus*) and a golden eagle (*Aquila chrysaetos*), and one species belonging to the order of the *Falconiformes*, a kestrel (*Falco tinnunculus*). Additionally, a pigeon (*Columba livia f. domestica*) belonging to the *Columbiformes* was examined. 

All animals were presented as patients in the Department of Small Mammal, Reptile and Avian Diseases, University of Veterinary Medicine Hannover, Foundation or sent over for necropsy. The ones presented as patients died or had to be euthanized because of severe diseases, and the owners provided them to the clinic for scientific purposes. 

The Ethics Committee of the University of Veterinary Medicine of Hannover, Foundation approved all procedures for obtaining samples for the analysis used in this study (ethic code: TVA-2020-V-28). The use of dead animals complied with German animal welfare laws, guidelines and policies; the study did not involve living animals.

The hearts were isolated from the body within 24 h after death and stored at −70 °C. Directly before the RNA isolation, equal-sized samples of the left and right atria and ventricles, about 10 to 15 mg each, were taken from the hearts. 

### 2.2. RNA Isolation and PCR

The RNA was isolated from the hearts using the MasterPureTM RNA Purification Kit (Epicentre by Illumina, San Diego, CA, USA), and the concentrations were measured after the isolation with the NanoDrop™ 1000 Spectrophotometer (Thermo Fisher Scientific Inc., Waltham, MA, USA). Concentrations showed a range of approximately 500 to 2500 ng/µL. Afterwards, the RNA was stored at −70 °C for up to a month. 

In preparation for the PCR, 8 µL of RNA were converted into cDNA using the SuperScript^®^ III First Strand Synthesis System for RT-PCR (InvitrogenTM by Thermo Fisher Scientific Inc., Waltham, MA, USA). Subsequently, a PCR was performed with a forward (5′-GAGAGGGACCTGAAGAGACAT-3′) and a reverse (5′-CGTCTTGGGAGGAACAGGTAC-3′) gene-specific primer, published and designed to detect BNP in pigeons [4] (Figure 1). 35 cycles of PCR were performed after an initial denaturation at 94 °C for 2 min, each cycle consisting of a denaturation at 94 °C for 30 s, annealing at 60 °C for 30 s and extension at 72 °C for 45 s. A final extension at 72 °C for 2 min was performed afterwards [4].

### 2.3. Gel Electrophoresis and Sequencing

The PCR products were run through a gel electrophoresis on a 2% agarose gel (Biozym Scientific GmbH, Hessisch Oldendorf, Germany) for 35 min at 120 V and 400 mA (Bio-Rad Laboratories GmbH, Feldkirchen, Germany; Figure 1). A DNA marker covering a range of 50 base pairs (bp) up to 2 kbp (Biozym Scientific GmbH, Hessisch Oldendorf, Germany) was used for visualization. The PCR bands were then cut out, cleaned up with the Nucleo-Spin^®^ Gel and PCR Clean-up (Macherery-Nagel GmbH & Co. KG, Düren, Germany), and bidirectionally sequenced in an external laboratory by Sanger Cycle Sequencing (Microsynth Seqlab GmbH, Göttingen, Germany).

### 2.4. Analysis

The sequencing results, both a forward and a reverse sequence for each examined species, were then aligned using BioEdit^®^ (Version 7.2.5) and translated into amino acid sequences. 

The Consensus sequences were then compared to other BNP sequences from bird species already examined in the past with GenBank^®^ and BioEdit^®^. The alignments were analyzed for their similarity percentages using EMBL-EBI services (EMBL-EBI^®^, Wellcome Genome Campus, Hinxton, Cambridgeshire, UK), and a phylogenetic tree was designed with Geneious Prime^®^ (Version 2021.0.1). 

## 3. Results

In the PCR products isolated from the hearts, the prepropeptide of BNP, including the signal sequence (72 bp), NT-pro-BNP (258 and 255 bp, respectively, see below) and the mature peptide region (90 bp), could be found. All PCR products consisted of approximately 480 bp in total. 

### 3.1. Mature Peptide Region

The nucleotide sequence of the mature peptide region, consisting of 90 nucleotides excluding the stop codon, was similar in all examined species, though substitutions of single nucleotides could be found (Figure 2). For example, all raptors, as well as the owls, showed substitutions of adenine to guanine and reverse in four different positions, including one within the stop codon, which were not shared by the parrots. The only exception were the three cockatoo species, which shared the substitution from guanine to arginine at the 42nd position. In addition, the cockatoo species shared substitutions of nucleotides in two other places that no other species showed. The budgerigar and eastern rosella also shared a substitution from adenine to guanine at the 83th position that could not be detected in any other species. The budgerigar also showed substitutions in two other positions, but those could not be found in the eastern rosella. Both lovebird species shared their complete nucleotide sequence with that of the congo grey parrot, whilst the scarlet macaw and blue-and-gold macaw shared a substitution from guanine to arginine at position number 27th. 

The amino acid sequence of the mature peptide region consisted of 29 amino acids and was concurrent in most examined species. All three owl species as well as the three raptors were identical. Deviations were only found in two of the parrot species, precisely the budgerigar and eastern rosella. Both species share a change from lysin to arginine at the 28th position, whilst the budgerigar also shows a substitution from asparagine to histidine at the 29th position (Figure 3). Compared to the mature peptide region of BNP of other species, all species except the budgerigar and eastern rosella were identical to the sequence known from the congo and timneh grey parrots as well as chickens [4,19]. There was a substitution of serine to proline at the 26th position in the pigeon’s sequence, which could not be found in any of the examined species (Figure 3). 

### 3.2. Signal Sequence and NT-pro-BNP

The preliminary sequence of the signal sequence and NT-pro-BNP shows a distinct resemblance through all species, although substitutions of single amino acids could be located throughout all of the examined species (Figure 3). All species examined in this study shared the same amino acids in 69% of the 140 positions, and the remaining 31% showed deviations in at least one species. Positions where it was not possible to define the exact amino acid in one of the species were calculated as possible substitutions. In the following, we will describe some of the most interesting substitutions, and all substitutions are summarized in Table 1. Table 2 shows the similarities between the complete prepropeptide sequences of all examined species. 

Compared to the rest the congo and timneh grey parrots showed the same differing amino acids in three positions (5th, 10th and 65th). Only position 22nd differed between the two species. 

Quite similar but in other positions, the Solomons cockatoo and citron-crested cockatoo shared a substitution in three positions that could not be found in most of the other species (3th, 8th and 91th). 

Both macaw species showed asparagine at the 59th position, whilst all other species showed lysin in the same position. The two lovebird species shared a substitution from glutamic acid to glycine in the 64th position and one from methionine to isoleucine in the 76th position. The former did not show up in any other species, whilst the raptors and two owls shared the latter. All owls shared a substitution from serine to asparagine in the 29th position that no other species shared. Additionally, the eagle owl and tawny owl shared asparagine in the 79th position, whilst all other species, including the barn owl, showed serine.

The amino acid sequence of the hooded vulture, as well as the one from the golden eagle, consisted of only 139 amino acids. All other species contained glutamic acid at position 70, but in those two species it was missing. 

When comparing the obtained sequence of the pigeon with the ones already known and published on GenBank^®^, substitutions of single amino acids stand out. Both pigeon sequences shared 99%. The differences appeared only in the sequence of the signal sequence and NT-pro-BNP, while the sequence of the mature peptide region was concurrent in both sequences (Figure 4).

Figure 5 shows a phylogenetic tree based on the similarities between the BNP prepropeptide of the examined species, in combination with the chicken, tortoise and crocodile. 

## 4. Discussion

The aim of the present study was to investigate the sequence of the suspected BNP with the prepropeptide in the heart of several bird species. Therefore, a primer originally designed to detect pigeon BNP was used, since earlier studies suggested that there was a high similarity between BNP in birds [4,5]. This perception could previously be supported by the examination of the BNP sequence of congo and timneh grey parrots, whose mature peptide region was identical to the ones known from chickens [4]. The present study extended the examined species to several more parrot species, some owls and birds of prey. The results also back the theory of a highly conserved mature peptide region of BNP in birds [5]. 

Interestingly, both species that showed a substitution in the amino acid sequence of the mature peptide region, the budgerigar and eastern rosella, are species originating from Australia. Other examined species that also originated from Australia did not show the same substitution, but they belonged to the family of the *Cacatuidae* in contrast to the family of *Psittacidae* [22]. 

The examination of the nucleotide sequence of the mature peptide region revealed interesting substitutions shared by species belonging to the same family, such as the substitutions shared by the cockatoo’s, raptors or owls. Other substitutions only appeared in a single species and could not be detected in other species, closely related or not. More studies with more animals of each species are needed to prove whether these substitutions are a species’ characteristic or just a particularity of the examined animal, as we already found in the timneh grey parrot (Figure 2). Three examined animals showed a substitution of one nucleotide when compared to each other, but none of them changed the amino acid sequence [21]. 

The examination of the signal sequence and NT-pro-BNP showed interesting joint substitutions of species belonging to the same family as well, such as the substitutions located in the macaws, lovebirds and cockatoos. However, great parts of the amino acid sequence were identical throughout all of the examined species. Since not all of the examined species belong to the same order of birds, one could suspect that the prepropeptide may be quite conserved in birds, even though obviously not as highly conserved as the mature peptide region. 

The deviation in the length of the amino acid sequence of the prepropeptide found in the golden eagle and hooded vulture requires further research, since a shortened sequence did not appear in any of the other species. The results should be confirmed with more animals of the same species and other species of the order *Accipitriformes*, which might show the same deviation. Interestingly, a length deviation could not be found in the other examined raptor, the kestrel. Because the kestrel belongs to another order, the *Falconiformes*, the BNP might have developed differently between these orders during evolution [23].

When compared to the already identified sequence of BNP of the pigeon, this study could only partly match the findings of prior studies. The sequence detected for the mature peptide region is concurrent with results from other studies, but we revealed some sequence differences for the prepropeptide. It remains unclear whether these differences are caused by breed differences, errors in the PCR process or the assay, or whether some of them might be caused by individual substitutions of the examined animal. Further studies with more animals of each examined species are necessary to determine the frequency of individual differences, which was not possible in this study since all animals were former patients of the clinic and the number of owners who provide their animals to science after death varies a lot.

This study detected BNP and NT-pro-BNP in the heart of several species that are relevant to avian consultation and in species that can be affected by cardiovascular diseases [24,25,26,27,28,29,30].

However, many species that are relevant for the avian veterinarian still remain to be examined, and further research is needed to investigate whether BNP or NT-pro-BNP might be a possible tool for diagnosing and monitoring cardiovascular diseases in birds, as it is in humans. Unfortunately, the mammal mature peptide region is quite different to the avian ones, since it differs much more between species and consists of 32 amino acids instead of 29 [5,21]. Therefore, tests invented for mammals will most likely not be applicable for birds. It was possible for the authors to detect BNP in the blood plasma of grey parrots with an ELISA originally designed for chickens for scientific purposes, and first results have been presented [31]. More studies are needed to examine whether BNP is detectable in the blood plasma of all the species in this study, what the physiologic concentrations are, which test method works best, and what factors have an impact on its concentration before conclusions can be made about whether BNP might be used as a diagnostic test in birds in the future.

## 5. Conclusions

In conclusion, this study detected BNP and NT-pro-BNP in the heart of 18 bird species belonging to the orders *Psittaciformes*, *Strigiformes*, *Accipitriformes* and *Falconiformes*. The results support the thesis of a highly conserved 29 amino acids long mature peptide region in birds, whilst the amino acid sequences of the prepropeptides showed some differences between the species. Further studies are necessary to develop a sufficient test system and evaluate the diagnostic potential of BNP and NT-pro-BNP for avian medicine. 

## Figures and Tables

**Figure 1 vetsci-09-00064-f001:**
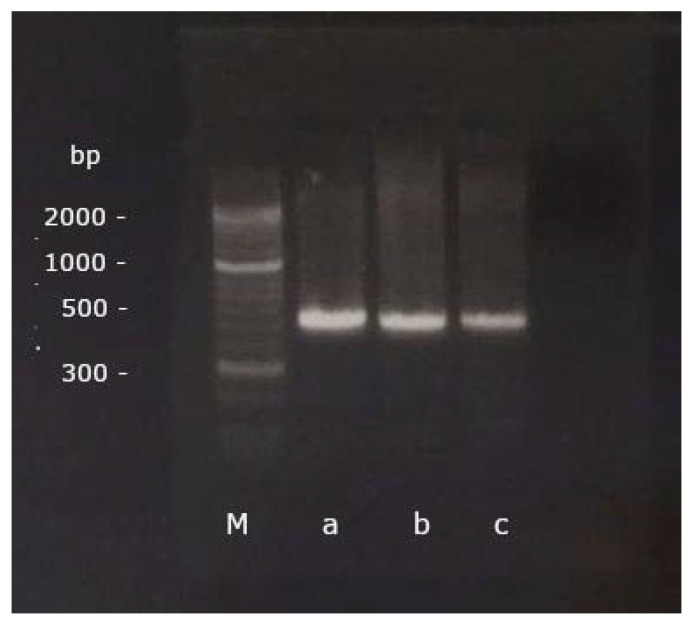
The PCR products of a grey parrot (**a**), a scarlet macaw (**b**) and a cockatiel (**c**) aligned with the marker (M) on the GelRed^®^-stained 2% agarose gel after gel electrophoresis (base pairs, bp).

**Figure 2 vetsci-09-00064-f002:**
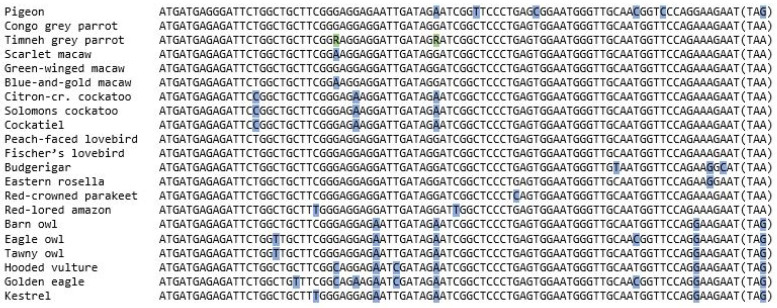
Nucleotide sequences of the mature peptide region of the BNP of the examined species aligned with the sequences of the pigeon and grey parrots, all consisting of 90 nucleotides. The stop codon is shown in brackets. The nucleotides that differ from the majority of the examined species are shaded in blue. Aberrations between animals of the same species as they were found in a previous study are shaded in green [21].

**Figure 3 vetsci-09-00064-f003:**
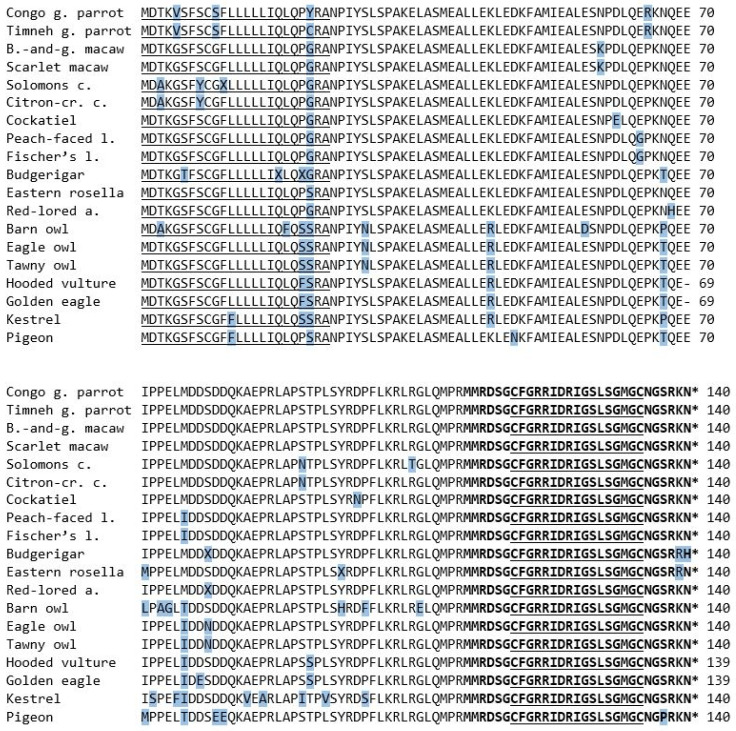
The amino acid sequences of the BNP prepropeptide of the species examined in this study. The 17-amino-acid ring structure is in bold letters and underlined. The asterisk marks a stop codon. The amino acids that differ from the majority of the examined species are shaded in blue.

**Figure 4 vetsci-09-00064-f004:**
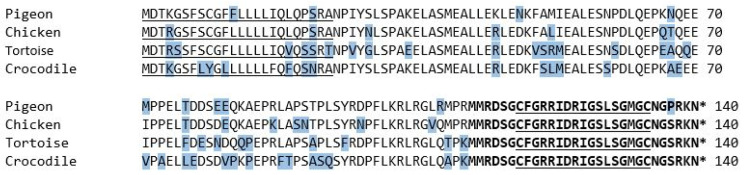
The amino acid sequence of the BNP prepropeptide of pigeons and chickens, as already known from previous studies. The signal peptide sequence is underlined, and the mature peptide region is shown in bold letters. The 17-amino-acid ring structure is in bold letters and underlined. The asterisk marks a stop codon. The amino acids that differ from the majority of the examined species are shaded in blue. GenBank accession numbers: NM_204925.1 *Gallus gallus* natriuretic peptide A (NPPA); NM_001282844.1 *Columba livia* natriuretic peptides A-like (BNP); AY398687.1 *Crocodylus porosus* B-type natriuretic peptide; AY433953.1 *Chelodina longicollis* B-type natriuretic peptide (BNP).

**Figure 5 vetsci-09-00064-f005:**
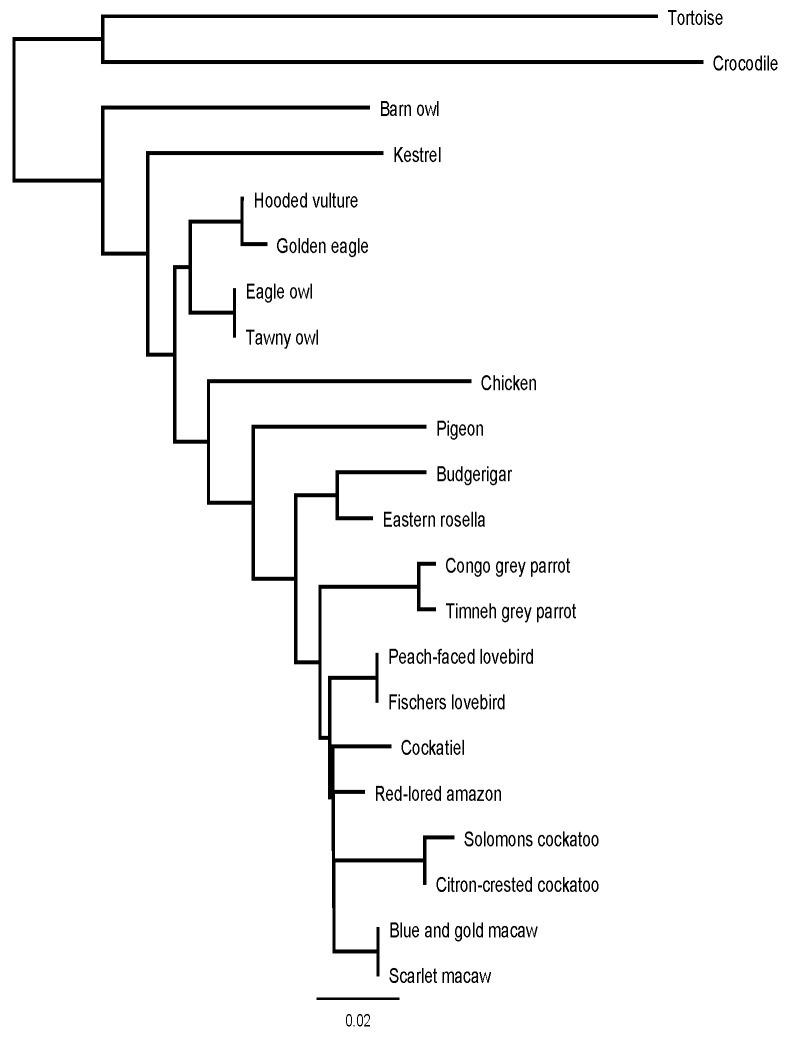
A phylogenetic tree of the examined species aligned with chicken, tortoise and crocodile sequences, based on the prepropeptide amino acid sequences from Figure 3 and Figure 4. The genetic distance in substitutions per position is shown below.

**Table 1 vetsci-09-00064-t001:** The positions with deviations in the amino acid sequence of the BNP prepropeptide from the majority of species examined in this study, as seen in Figure 3.

Species	Position of Substitution
Congo grey parrot	5, 10, 22, 65
Timneh grey parrot	5, 10, 22, 65
Blue and gold macaw	59
Scarlet macaw	59
Solomons cockatoo	3, 8, 91, 105
Citron-cr. Cockatoo	3, 8, 91
Cockatiel	61, 98
Peach-faced lovebird	64, 76
Fischer’s lovebird	64, 76
Budgerigar	6, 67, 139, 140
Eastern rosella	22, 71, 139
Red lored amazon	68
Barn owl	3, 19, 21, 22, 29, 45, 57, 67, 71, 73, 74, 76, 96, 99, 106
Eagle owl	21, 22, 29, 45, 67, 76, 79
Tawny owl	21, 22, 29, 45, 67, 76, 79
Hooded vulture	21, 22, 45, 67, 76, 92
Golden eagle	21, 22, 45, 67, 76, 78, 92
Kestrel	21, 22, 45, 67, 72, 75, 76, 84, 86, 91, 94, 99
Pigeon	22, 48, 67, 71, 76, 80, 81, 137

**Table 2 vetsci-09-00064-t002:** The similarities, in percentages, of the BNP prepropeptide of the species examined in this study and the sequence of the chicken GenBank accession number: NM_204925.1 *Gallus gallus* natriuretic peptide A (NPPA).

	Congo Grey Parrot	Timneh Grey Parrot	Blue and Gold Macaw	Scarlet Macaw	Solomons Cockatoo	Citron-cr. Cockatoo	Cockatiel	Peach-Faced Lovebird	Fischer’s Lovebird	Budgerigar	Eastern Rosella	Red-Lored Amazon	Barn Owl	Eagle Owl	Tawny Owl	Hooded Vulture	Golden Eagle	Kestrel	Pigeon	Chicken
Congo g. parrot Timneh g. parrot B.-and-g. macaw Scarlet macaw Solomons c. Citron-cr. c. Cockatiel Peach-faced l. Fischer’s l. Budgerigar Eastern rosella Red-lored amazon Barn owl Eagle owl Tawny owl Hooded vulture Golden eagle Kestrel Pigeon Chicken	100.00	99.29 100.00	96.43 96.43 100.00	96.43 96.43 100.00 100.00	93.57 93.57 95.71 95.71 100.00	95.00 95.00 97.14 97.14 98.57 100.00	95.71 95.71 97.86 97.86 95.00 96.43 100.00	95.71 95.71 97.86 97.86 95.00 96.43 97.14 100.00	95.71 95.71 97.86 97.86 95.00 96.43 97.14 100.00 100.00	92.14 92.14 94.29 94.29 91.43 92.86 93.57 93.57 93.57 100.00	95.00 95.00 96.43 96.43 93.57 95.00 95.71 95.71 95.71 93.57 100.00	95.71 95.71 97.86 97.86 95.00 96.43 97.14 97.14 97.14 95.00 95.71 100.00	87.14 87.14 88.57 88.57 87.14 88.57 87.86 88.57 88.57 85.71 89.29 87.86 100.00	92.86 92.86 94.29 94.29 91.43 92.86 93.57 95.00 95.00 92.86 93.57 94.29 91.43 100.00	92.86 92.86 94.29 94.29 91.43 92.86 93.57 95.00 95.00 92.86 93.57 94.29 91.43 100.00 100.00	93.53 93.53 94.96 94.96 92.09 93.53 94.24 95.68 95.68 92.81 94.24 94.24 89.93 97.12 97.12 100.00	92.81 92.81 94.24 94.24 91.37 92.81 93.53 94.96 94.96 92.09 93.53 93.53 89.21 96.40 96.40 99.28 100.00	88.57 88.57 90.00 90.00 87.86 89.29 89.29 90.71 90.71 87.14 89.29 89.29 87.14 92.14 92.14 92.09 91.37 100.00	91.43 91.43 92.86 92.86 90.00 91.43 92.14 92.86 92.86 90.00 93.57 92.14 87.14 92.14 92.14 92.81 92.09 88.57 100.00	87.86 87.86 89.29 89.29 87.86 89.29 90.00 89.29 89.29 86.43 88.57 88.57 85.71 91.43 91.43 90.65 89.93 85.71 89.29 100.00

## Data Availability

Data is contained within the article.

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
