# Peer review of "Analysis of the B-Type Natriuretic Peptide and the Aminoterminal-Pro-B-Type Natriuretic Peptide in Different Parrot, Raptor and Owl Species"

_vetsci, 2022, doi:10.3390/vetsci9020064_

Round 1

Reviewer 1 Report

The manuscript deals with the sequencing of B-Type Natriuretic Peptide (BNP) and its prepropeptide, NT-pro-BNP isolated from different exotic and wild avian species. The study design is fair, the applied methods are correctly described, and the manuscript is clearly written with fine scientific English and with very few spelling mistakes. However, I have some comments and concerns, which should be addressed before being considered for publication:

  1. In the frame of the study, BNP and NT-pro-BNP are newly sequenced from 13 parrot, 3 raptor and 3 owl species. In the title, only "...in birds" is indicated, which is not correct as only certain exotic/wild bird species are investigated, so please modify it to be more accurate and more specific.
  2. The authors present the sequence data and discuss the most important differences in the text. However, it would be important and valuable to provide some further quantitative analyses and graphical presentation concerning similarities and dissimilarities between the amino acid sequences of different species. For instance, the paper of Abo-Elkhier et al., 2019 (https://www.hindawi.com/journals/bmri/2019/2796971/) presents several such possibilities for quantitative comparisons.
  3. Please present some representative images of the PCR products after gel electrophoresis (for the mature BNP peptide, NT-pro-BNP and the signal sequence).
  4. In my opinion, assessing the nucleotide and amino acid sequence of BNP and NT-pro-BNP in different avian species is important by providing valuable basic science data; however, it would have more merit to study if BNP/NT-pro-BNP can be applied as a marker of cardiovascular diseases in birds. Describing the sequence alone in several species without functional/physiological studies is not of high scientific interest for the clinical veterinary medicine, being a weakness of the present study. This is the reason why I have indicated "average" for the significance/scientific soundness/merit of the manuscript.
  5. Are there any analytical methods for detecting BNP/NT-pro-BNP in the blood plasma of birds? Can available ELISA kits (targeting mammalian species) be applied for birds as well? How is the similarity of the amino acid sequence between mammals and birds? The existence or  the required development of analytical methods is an important pre-requisite of future studies regarding the clinical significance of BNP/NT-pro-BNP as diagnostic/prognostic markers. Please comment on this in the manuscript.
  6. In Figure 1, it is difficult to differentiate light and dark grey shading, please consider using another highlighting method for intra-/interspecific differences.
  7. In the caption of Figure 2, there is probably a spelling mistake: "Table 17. amino-acid ring structure..." is written, I guess the authors meant "The 17-amino-acid ring structure...".

Reviewer 2 Report

This is a well-written manuscript documenting an important discovery, i.e. the amino acid sequences of BNP in various bird species. However, the conclusions of its use as a clinical diagnostic tool are a bit exaggerated and a more realistic conclusion is required. The results can also be presented in a clearer manner.

Introduction:

The introduction is well-written and summarizes the current knowledge regarding BNP in birds very well.

Materials and methods:

The materials and methods sections is well-described, allowing easy replication of the study. A good range of species were represented in the study, although the lack of duplication for each species should be justified in the discussion.

Results:

Line 163: Replace "22th" with "22nd".

Line 154-184: This entire section could potentially be better summarized in a table, with the species in one column and position of substitution in another.

Figures 1, 2 and 3: The highlighted positions will be clearer in color instead of grayscale.

Discussion:

Line 246-251: The detection of amino acid sequences corresponding to BNP in various bird species is a big step towards its utilization as a clinical diagnostic tool, but is also a few steps removed from the actualization. These missing steps should be pointed out in the conclusion as steps for further studies, e.g. detection of BNP itself in heart tissues, detection of BNP in the bloodstream, establishment of blood BNP reference ranges in healthy birds etc.
